# Adaptive Impedance Matching Network for Contactless Power and Data Transfer in E-Textiles

**DOI:** 10.3390/s23062943

**Published:** 2023-03-08

**Authors:** Pim Lindeman, Annemarijn Steijlen, Jeroen Bastemeijer, Andre Bossche

**Affiliations:** Faculty of Electronic Engineering, Mathematics and Computer Science, Delft University of Technology, 2628 CD Delft, The Netherlands

**Keywords:** compensation networks, data transmission, e-textiles, inductive coupling, resonant circuit, wireless sensor networks, wireless power transfer

## Abstract

One of the major challenges associated with e-textiles is the connection between flexible fabric-integrated wires and rigid electronics. This work aims to increase the user experience and mechanical reliability of these connections by foregoing conventional galvanic connections in favor of inductively coupled coils. The new design allows for some movement between the electronics and the wires, and it relieves the mechanical strain. Two pairs of coupled coils continuously transmit power and bidirectional data across two air gaps of a few millimeters. A detailed analysis of this double inductive link and associated compensation network is presented, and the sensitivity of the network to changing conditions is explored. A proof of principle is built that demonstrates the system’s ability to self-tune based on the current–voltage phase relation. A demonstration combining 8.5 kbit/s of data transfer with a power output of 62 mW DC is presented, and the hardware is shown to support data rates of up to 240 kbit/s. This is a significant improvement of the performance of previously presented designs.

## 1. Introduction

The increasing miniaturisation and power efficiency of integrated circuits has allowed for electronic devices to be more easily integrated into everyday items. The clothing industry has not been blind to this trend and the interest in and market for electronics integrated into clothing, known as e-textiles, is growing [1,2].

The most common use of e-textiles is the measurement of certain body parameters. For the sensing of body-wide metrics, such as body temperature [3], heart rate [4,5], breath rate [6], and blood pressure [7], a single node is sufficient. However, the measurement of certain localized metrics, such as muscle activity [8] or limb movement [9], require a network of sensors distributed over the garment. For reasons of cost, efficiency, and spectral pollution, it is not desirable to have each node in such a network wirelessly connected to the outside world. It is preferred to have all the sensor nodes within the garment connected to a central node. This central node will communicate with the outside world, and supply power to all other nodes. Such a system is known as a body area network (BAN).

Most implementations of a BAN involve connecting the nodes using conductive paths that are directly integrated into the clothing [10,11]. These interconnecting paths carry data from, and supply power to the sensors, ideally without compromising on the comfortability and wearability of the garment. After many years of development, conductive paths in clothing can now be made to withstand the mechanical stresses of everyday wear [12].

However, these conductive paths need to be connected to the (usually rigid) electronics. This can be done mechanically [13,14], by way of soldering [15,16], polymer adhesion [17], or PCB embroidering [18]. However, concerns have been raised over the long term reliability [19,20] as well as the end-of-life care of garments with electronics permanently embedded [21].

This work aims to investigate if it is viable to replace the galvanic connections with coupled coils, eliminating the risk of mechanical contact failure, and making the placement and removal of the electronics more user-friendly. This method will allow communication between nodes placed anywhere inside a garment, regardless of distance. A schematic representation of the system is shown in Figure 1.

The use of an inductive link to transfer both power and data is by no means new [22]. The two most prominent applications of this technique are currently electric vehicle charging and medical implants. These two applications are very different in scope and technique. The first has allowed 3.3 kW of power transfer to be combined with up to 64 kbits/s of data transfer [23], while the latter allows for power transfer in the mW range, combined with data speeds measured in Mbits/s [24].

Some research into the use of coupled coils in textiles has been presented recently. Attempts have been made to combine off-the-shelf near-field communication (NFC) technology with embroidered antennas to form contactless connections [25,26,27,28]. The idea of creating full body-wide BANs using multiple antennas with in-fabric interconnects has been presented before [29,30,31,32]. However, these publications focused either on the performance of the coils and did not implement a full system, or they focused on the full system and did not provide any data on the achieved power and data transfer across the link.

Of the most relevance to our work is the system created by Lin et al. [29]. They created a design for a BAN using inductively coupled sensors for posture and temperature monitoring with sensor nodes positioned on various points on the body. These sensors would inductively couple to coils embedded into a flexible polyester-spandex shirt. Coils and interconnects were embroidered directly into the fabric of the garment using conductive threads. The system relied on off-the-shelf NFC chips to facilitate the data and power transfer. A limited data transfer rate of 112 bits/s was demonstrated, combined with a total transferred power of 12 mW at an efficiency of 6%.

The goal of our work is to much improve upon the data and power transfer rates achieved in [29], with data rates in the tens of kilobits. The system is also made to be resilient to changing conditions, particularly changing parameters caused by movement of the coils. To achieve this, our system does not rely on any preexisting NFC protocol, as these are not designed to transfer a significant amount of continuous power and data. Instead, both the system hardware and software are tailor made to suit the application.

This publication is based on a master thesis on the same topic [33]. Aside from the information presented in the thesis, this article contains a detailed analysis of the system sensitivity to changing link conditions.

The design is divided into three parts: the master, the slave, and the link. The master interfaces with the battery and the transmitter on one side, and the link inside the garment on the other. It is responsible for generating the carrier signal, and modulates this signal to send data and acknowledgments to the slave. It also demodulates the data coming from the slave and presents this in a usable form to the transmitter. The slave acts as a bridge between the link and the sensor. It harvests energy from the carrier and performs demodulation. It also modulates the data from the sensor onto the carrier by way of load modulation.

The link forms the connection between the master and the slave. It consists of four planar coils, forming two pairs, as well as some surrounding capacitors. A diagram of these parts, as well as the subsystems within them, is shown in Figure 2.

First, the link and accompanying compensation circuit are analyzed. In the second section, the design of the subsystems comprising the slave and the master is presented. Third, the practical implementation is shown, and the measurements are addressed in the fourth section. Finally, the conclusions drawn from this work are presented.

## 2. Methods

### 2.1. Link Design

In this section, the design and characteristics of the link and the associated compensation circuits are discussed. The purpose of the link is to carry power to the sensor node, and to allow for bi-directional communication between the central node (master) and the sensor node (slave).

#### 2.1.1. Link Characterization

In order to design a compensation network, the link is modeled as a two-port system. The equivalent circuit used for the model consists of two pairs of coupled coils with the non-ideal coupling factors $k_{1}$ and $k_{2}$. Any parasitic capacitances and resistances are not considered at this time. The equivalent circuit used for the analysis is shown in Figure 3.

Here, $L_{11}$ through $L_{22}$ indicate the self inductance of the coils, and ω is the signal frequency. To find the transmission matrix of this network, first, the transmission matrix of a single link (Figure 4) is analyzed:(1)$$\left[\begin{array}{c} V_{in}\\ I_{in} \end{array}\right]=\frac{1}{\hat{k}\sqrt{\hat{L_{1}}\hat{L_{2}}}}\left[\begin{array}{cc} \hat{L_{1}} & j\omega \hat{L_{1}}\hat{L_{2}}\left(1-{\hat{k}}^{2}\right)\\ \frac{1}{j\omega } & \hat{L_{2}} \end{array}\right]\left[\begin{array}{c} V_{out}\\ I_{out} \end{array}\right]$$

The transmission matrix of the double link can be found by multiplying the transmission matrices of the two single links as shown in (Equation 2).
(2)$$\left[\begin{array}{c} V_{in}\\ I_{in} \end{array}\right]=\frac{1}{k_{1}k_{2}\sqrt{L_{11}L_{12}L_{21}L_{22}}}\left[\begin{array}{cc} L_{11}L_{21}-L_{11}L_{12}\left(k_{1}^{2}-1\right) & j\omega L_{11}L_{22}\left[L_{12}\left(1-k_{1}^{2}\right)+L_{21}\left(1-k_{2}^{2}\right)\right]\\ \frac{L_{12}+L_{21}}{j\omega } & L_{21}L_{22}-L_{21}L_{22}\left(k_{2}^{2}-1\right) \end{array}\right]\left[\begin{array}{c} V_{out}\\ I_{out} \end{array}\right]$$

In order to simplify the equations, the following assumptions are made. First, it is assumed that the coils used on the slave and master side electronics are of the same design, meaning that $L_{11}=L_{22}=L$. The same assumption is made about the coils that would be implemented in the textile: $L_{12}=L_{21}=L_{2}$. Finally, it is assumed that the mounting of the coil pairs on both sides is the same; ergo, $k_{1}=k_{2}=k$. This simplifies (Equation 2) to:(3)$$\left[\begin{array}{c} V_{in}\\ I_{in} \end{array}\right]=\frac{1}{k^{2}}\left[\begin{array}{cc} 2-k^{2} & 2j\omega L(1-k^{2})\\ \frac{2}{j\omega Lk} & 2-k^{2} \end{array}\right]\left[\begin{array}{c} V_{out}\\ I_{out} \end{array}\right]$$
Note that (Equation 3) does not contain $L_{2}$, meaning that, under these assumptions, the self inductance of the inner coils does not play a role. Furthermore, applying the following substitution:(4)$$\hat{L}=\frac{1}{2}L\left(2-k^{2}\right),\;\hat{k}=\frac{k^{2}}{2-k^{2}}$$
to (Equation 1) yields (Equation 3). This shows that, using these assumptions, the double link behaves the same as a single link with an applied transformation.

#### 2.1.2. Link Compensation

For effective power transfer, the impedance at each interface needs to be matched. However, since the signal generator in the slave and the rectifier in the load are non-linear in nature, the exact values of these impedances are ill-defined. Moreover, the input impedance of the slave is dependent on its power draw, which cannot be guaranteed to be constant over time. In order to disconnect the design of the link from that of the source and load, the link is designed in such a way that the impedance of the load is mirrored to the source. In this way, the inductive link essentially behaves as a direct wired connection at the operating frequency. This ideal behavior will result in a system transmission matrix equal to the identity matrix.

The matching of impedances in a WPT (wireless power transfer) system is commonly achieved by placing capacitors in series or parallel with the link. This is known as link compensation or resonant WPT [34,35,36,37,38]. In the last section, it was shown that the double link can be modeled using a conventional single link, meaning that it is possible to use the same compensation techniques in a double-link network as are used in single-link networks. Although others exist and are used, the most common compensation techniques are SS (series–series), SP (series–parallel), PS (parallel–series), and PP (parallel–parallel) [34].

In this naming convention, the first and second words refer to capacitors added to the primary and secondary side, respectively. The values of these capacitors are chosen in such a way that, when the system is loaded with a resistive load, the system is resonant at the operating frequency [38]. The component values, as well as the resulting transmission matrix when applying the system to the single link in Figure 4, are shown in Table 1. The system is analyzed at the operating frequency $\omega_{0}$.

From this table, it can be seen that the transmission matrices of the SS and PS will not converge to the identity matrix. The PP and SP will converge when $\hat{k}$ approaches 1; however, it is assumed that this is not attainable in most cases. Moreover, *k* is likely to vary over time, as the coils move with respect to each other. Since both $\hat{L}$ and $\hat{k}$ are dependent on *k* in the double link, both $C_{p}$ and $C_{s}$ would need to vary in order to maintain resonance.

For this reason, a new three capacitor compensation network is proposed. In this network, only a single capacitor value is dependent on the coupling *k*. A schematic representation of this is shown in Figure 5. The component values are:(5)$$C_{p}=C_{s}=\frac{1}{\omega_{0}^{2}L}$$
(6)$$C_{comp}=\frac{{\bar{k}}^{2}}{2L\omega_{0}^{2}\left(1-{\bar{k}}^{2}\right)}$$

Here, $\bar{k}$ is an estimate of the actual coupling factor *k*. From Equations (Equation 5) and (Equation 6), it can be seen that $C_{s}$ and $C_{p}$ form a resonant pair with the uncoupled self inductance of *L* at the operating frequency. Only $C_{comp}$ is dependent on the coupling factor. The resulting transmission matrix of this network is shown in (Equation 7). As $\bar{k}$ approaches *k*, this matrix will converge to the identity matrix.
(7)$$\left[\begin{array}{c} V_{in}\\ I_{in} \end{array}\right]=\left[\begin{array}{cc} 1 & -j\omega_{0}\hat{L}\frac{{\hat{k}}^{2}\bar{k}-\hat{k}{\bar{k}}^{2}+\hat{k}-\bar{k}}{\hat{k}\bar{k}}\\ 0 & 1 \end{array}\right]\left[\begin{array}{c} V_{out}\\ I_{out} \end{array}\right]$$

#### 2.1.3. Quality Factor

Since the link needs to carry not only power but also data, it is important to investigate its dynamic behavior. To this end, the quality factor *Q* of the system is assessed. The quality factor determines the settling time of the system, and places an upper bound on the speed at which data can be transferred. The quality factor of a system is defined as the ratio between the stored power and the dissipated power [39]. For fast data transfer, the quality factor needs to be as low as possible. In practice, *Q* can be expressed as:(8)$$Q=\frac{\omega E_{s}}{P_{dis}}$$
where $E_{S}$ is the average energy stored in the circuit, and $P_{dis}$ is the average power dissipated. In this section, *Q* will be assessed under absolute ideal circumstances. All components are assumed lossless and do not have any parasitic capacitance associated with them. The network is assumed to be perfectly tuned ($\bar{k}=k$), and the signal is applied at the resonant frequency ($\omega =\omega_{0}$). The system is loaded with a fully real load resistance $R_{l}$.

In order to calculate the quality factor of the circuit, the average energy stored $E_{s}$ needs to be determined. This is simply the sum of the average energy stored in the capacitors $E_{C}$ and the inductors $E_{L}$.
(9)$$E_{s}=\sum E_{C}+\sum E_{L}$$
All the reactive elements in the circuit are in resonance, indicating that all the energy stored in the capacitors will be exactly delivered to the inductances in each cycle. This simplifies Equation (Equation 9) to:(10)$$E_{s}=2\sum E_{C}$$
Using conventional circuit analysis, the average energy in each capacitor can be determined by first calculating the voltage amplitude across each capacitor:(11)$$\begin{array}{cccc} \; & V_{Cp} & = & \;V_{in}\sqrt{1+\frac{1}{\omega^{2}C_{comp,A}^{2}R_{l}^{2}}}\\ \; & V_{Cs} & = & \;V_{in}\\ \; & V_{Ccomp} & = & \;V_{in}\frac{1}{\omega C_{comp,A}R_{l}} \end{array}$$
Using these voltages, the mean stored energy per component can be calculated using (Equation 12).
(12)$$E_{s}=\frac{1}{2}CV_{C}^{2}$$
Combining Equations (Equation 10)–(Equation 12) gives the following expressions for the energy stored in the circuit:(13)$$E_{s}=\frac{1}{2}V_{in}^{2}\left[\left(C_{p}+\frac{C_{p}}{\omega^{2}C_{comp}^{2}R_{L}^{2}}\right)+C_{s}+\frac{1}{\omega^{2}C_{comp}R_{l}^{2}}\right]$$
Inserting the values for $C_{p}$, $C_{s}$ and $C_{comp}$ from Equations (Equation 5) and (Equation 6) gives:(14)$$E_{s}=\frac{1}{2}V_{in}^{2}\left(\frac{L(1-k^{2})}{R_{l}^{2}k^{2}}+\frac{2}{L\omega^{2}\left(k+1\right)}\right)$$
As explained earlier, the input voltage is the same as the output voltage, and no power is assumed to be lost in the reactive components. It can thus be stated that:(15)$$P_{dis}=\frac{V_{in}^{2}}{2R_{l}}$$
Combining Equations (Equation 14) and (Equation 15) with the definition of the quality factor given by (Equation 8) results in an equation for the quality factor:(16)$$Q=\frac{\omega L}{R_{l}}\frac{1-k^{2}}{k}+\frac{R_{l}}{\omega L}\frac{2}{k(k+1)}$$

In Figure 6, the behavior of this equation is plotted for a typical system. When the coupling is ideal, a lower load will always result in a lower *Q*. When as the coupling decreases, the minimum achievable quality factor also decreases. For non-ideal coupling, there exists an $R_{l}$ that results in optimal performance. This optimal value of $R_{l}$ is dependent on *k*.

#### 2.1.4. Non-Idealities

In Section 2.1.2, it is demonstrated that, when optimally compensated, the load at the output to the link is reflected to the input without transformation. However, this property only holds as long as all of the components are sized perfectly. This cannot be guaranteed in practice, particularly if the coils are placed on a flexible substrate. These variations in component values will result in the link going out of resonance, which will manifest itself as a shift in the input impedance.

In order to quantify the effects of these variations, the input impedance for various mismatches are shown in Figure 7. Here, each column represents a parameter, which is varied from −10% to +10% of its nominal value. To indicate the effect of the coil self inductance $L_{xx}$ on the sensitivity, the rows correspond to different coil sizes. Each entry shows how the real and imaginary parts of the input impedance vary as a result of variations in the selected parameter.

There are a few conclusions that can be drawn from Figure 7. First, errors in the value of $L_{11}$ result in the largest deviations in input impedance, regardless of the size of the inductor used. Due to this large effect on the link, it is recommended that this coil is made from a rigid material even in wearable applications. This will limit the variations in self-inductance that result from deformation.

Variations in the coupling factors $k_{1}$ and $k_{2}$ also affect the system considerably, $k_{1}$ slightly more than $k_{2}$. However, these variations mostly result in an imaginary component being added to the link input impedance, rather than a transformation of the real component. An alteration in the imaginary part of the impedance can easily be compensated for by varying $C_{comp}$.

Finally, it can be noted that, in most cases, small self-inductance coils make the system more resilient to mismatches. The only exception to this is variations in $L_{22}$, where the opposite is true. However, the overall effect of $L_{22}$ on the system is very limited. To improve robustness to mismatch, it is therefore recommended to use coils with a small number of turns.

In addition to perfect component values, the theoretical analysis of the previous section also assumes that the distance between $L_{12}$ and $L_{21}$ is zero. In practice, this will not be the case. Any wire between the coils will act as a transmission line, which will introduce a voltage transformation as well as a phase shift, thereby, altering the input impedance. The effect of this is demonstrated in Figure 8. Low-impedance transmission lines are shown to introduce smaller absolute errors; however, high impedance lines almost exclusively add to the imaginary part of the impedance. This can, again, be easily corrected by varying $C_{comp}$.

#### 2.1.5. Practical Implementation

The coils comprising the link were made of rigid materials to simplify manufacturing and to increase consistency in the proof of concept. Furthermore, the wire connecting $L_{12}$ and $L_{21}$ was about 4 cm to minimize any transmission line effects. The coils were made using standard PCB technology with a copper thickness of 35 μm and a trace width of 4.5 mm. They consisted of five turns each and had an outside diameter of 5 cm. Their self inductance was measured at about 488 nH, and each pair had a coupling factor of roughly 0.72 when placed at a distance of 1.5 mm. They were brought in resonance at 13.56 MHz with a tunable capacitor. $C_{comp}$ was implemented as a voltage-controlled capacitor ranging from 100 to 200 pF.

### 2.2. Supporting Electronics

In this section, the circuitry required to send power and data over the link is discussed. Data are sent from the master to the slave using on-off keying (OOK), and data are sent back using load modulation. A 13.56 MHz carrier signal is generated by the master, and the slave uses this to harvest energy.

#### 2.2.1. Carrier Generator

Common to many other WPT systems, a class-E amplifier is used to generate the carrier signal [40,41,42,43,44,45]. However, the conventional class-E topology [46] was modified to better suit the application. There is typically a resonant tank placed at the output of the amplifier to suppress the higher order harmonics. This resonant tank will increase the system quality factor and, in turn, reduce the system’s ability to transmit data. Moreover, the link and accompanying compensation network is already a resonant network, and so higher order harmonics are already being blocked. Therefore, the resonant tank has been removed. The modified network is shown in Figure 9.

In order to maintain ZVS (zero-voltage switching), $L_{shunt}$ was significantly reduced in size from the conventional network. It works with $C_{par}$ to create a resonant network. This resonant network was tuned to allow for a single peak during the time when the clock signal is low. Therefore, the tuning of this network is highly dependent on the clock frequency and duty cycle. The unloaded voltage signal is given by:(17)$$\begin{array}{ccccc} \; & V(t) & \;=\; & Asin\left(\omega_{amp}\left(t-\tau \right)\right)+V_{DD}, & 0<t<DT\\ \; & V(t) & \;=\; & 0, & DT<t<T \end{array}$$
Here, $\omega_{amp}$ is the natural resonant frequency of $L_{shunt}$ and $C_{par}$, *A* is the amplitude of the sinewave given by (Equation 18), τ is a phase shift that ensures V(0)=V(T), and *T* and *D* are the clock period and duty cycle, respectively.
(18)$$A=\frac{\omega_{amp}\left(1-D\right)TV_{DD}}{cos\left(\omega_{amp}\tau \right)-cos\left(\omega_{amp}\left(DT-\tau \right)\right)}$$

In order to achieve ZVS, $\omega_{amp}$ should be:(19)$$\omega_{amp}=\frac{-2arcsin\left(\frac{V_{DD}}{A}\right)-\frac{\pi }{2}}{DT}$$

A plot of the resulting voltage waveforms and their spectral components is shown in Figure 10. From these plots, it can be seen that, as the duty cycle increases, the waveform approaches a pure sine wave, and the spectral efficiency improves.

The above equations describe the carrier generator under no-load conditions. When a load is applied, the quality factor of the resonant system formed by $L_{shunt}$ and $C_{par}$ is reduced, and the resonance is dampened. This will cause the model to no longer be accurate, and the system can start to violate the ZVS condition. Simulations show that, at a duty cycle of 50%, a quality factor of 6 will suffice. At a duty cycle approaching 100%, a quality factor of at least 25 is required for the model to remain valid. In order to increase the quality factor, it is recommended to choose a small $L_{shunt}$ paired with a large $C_{par}$.

The prototype system has a clock period of 74 ns and duty cycle of 65% combined with an $L_{shunt}$ of approximately 580 nH and a $C_{par}$ of 195 pF.

#### 2.2.2. Modulation and Demodulation

Communication from the slave to the master happens via load modulation. For this, a load modulator is embedded in the design of the rectifier used for energy harvesting. This is shown in Figure 11a. The advantage of this design is that, when the MOSFET is on, $R_{mod}$ is seen as a linear resistance in parallel with the rectifier. Some other designs would have $R_{mod}$ placed at the output of the rectifier, thereby, resulting in non-linear modulation. This design also eliminates any problems related to the body diode of the MOSFET. The value of $R_{mod}$ determines the modulation depth.

The modulation of the load can be detected on the master side by measuring the current flowing into the link. For this, a current-measuring circuit is implemented. To measure the current with a minimal impact on the performance, a transformer-based circuit is used combined with a current-to-voltage amplifier. The circuit for this is shown in Figure 11b. The primary side of the transformer is connected in series with the transmitting coil. This induces a current in the secondary side, which is amplified by the op-amp. The output voltage of this detector is given by:(20)$$V_{sense}=I_{coil}\frac{-j\omega k_{sense}\sqrt{L_{p}L_{s}}A_{v}R_{fb}}{R_{fb}+j\omega L_{s}\left(A_{v}+1\right)}$$
where $k_{sense}$ is the transformer coupling, $L_{p}$ and $L_{s}$ are the primary and secondary self inductance, $A_{v}$ is the op-amp voltage gain, $I_{coil}$ is the to-be-measured current, and $R_{fb}$ is the op-amp feedback resistance. As the op-amp gain increases, this converges to:(21)$$V_{sense}=-I_{coil}R_{fb}k_{sense}\sqrt{\frac{L_{p}}{L_{s}}}$$

The output signal is fed to an AD8302 RF magnitude detector, after which, some further analog processing is used to distinguish between the bits. This results in a digital bit stream.

Communication from master to slave is implemented using OOK. The varying carrier amplitude is decoded by a magnitude detector at the input of the slave. This arrangement results in a half-duplex system.

#### 2.2.3. Coupling Detection

Aside from demodulation, the current sense network described in the previous section can also be used to optimize the link compensation to the current coupling factor. This can be useful in situations where the coils can move with respect to each other, thus, causing *k* to vary over time. If the load is fully real, a perfectly coupled link will have a fully real input impedance. If there is an imaginary component to the link input impedance, this means that the link is not optimally compensated. By varying $C_{comp}$, optimal performance can be restored. By comparing the phase of the measured current signal with the voltage over the coil, the phase of the link input impedance can be measured, which can then be used to optimize $C_{comp}$.

#### 2.2.4. Practical Implementation

The system as designed was built using off-the-shelf components and standard PCB manufacturing techniques. The master and the slave were housed on separate boards. The boards were not optimized for size, and contained many test points for verifying each subsystem individually. A picture of the boards used in a test setup is shown in Figure 12.

## 3. Results and Discussion

### 3.1. Maximum Data Speeds

In order to test the maximum data rates that the system is capable of transmitting, a square wave was applied to the data input on both sides. This square wave was increased in frequency until the demodulated output signal no longer resembled the input signal. A signal was considered acceptable if the duty cycle of the demodulated signal was within a 0.35–0.65 range to allow the readout electronics some time to register the signal.

We found that communication from the master to the slave reached a maximum at 120 kHz or 240 kbits/s, and communication from the slave to the master was able to reach, at most, 180 kHz or 360 kbits/s. The measured waveforms at these frequencies are shown in Figure 13.

It can be noted from the waveforms that there is a delay between edges in the sent and received data. This delay is not consistent between rising and falling edges. In Figure 13a, the rising edge delay is shorter, which results in an elongation of the received pulse. The opposite is true in Figure 13b. This appears to be an artifact of the RF detectors used. It ultimately limits the data rate in this system, as increasing the frequency further would cause the pulse to become so short it collapses.

### 3.2. Power Transfer Capabilities

In order to determine the efficiency and power draw of the various subsystems, the system was measured in three different configurations. First, the system efficiency to an AC load was measured. Here, the entire slave was replaced by a load resistor. The input power was determined by measuring the DC voltage and current supplied to the master. The output power was the RMS power measured over the resistor. This allowed us to estimate the efficiency of the link, the carrier generator, and the master side demodulation electronics.

In the second measurement, the efficiency was measured for the DC load. Here, the load resistance was placed after the rectifier shown in Figure 11a. This provided an indication of the energy loss in the rectifier.

Finally, the entire slave was reconnected. The load resistance was placed at the 3.3 V output of the slave, where it simulated power being drawn by an external sensor. This enabled measurement of the full system efficiency, including the losses incurred in the output DC-DC converter, as well as the power required by the demodulation electronics in the slave.

All of the above experiments were performed without data transfer with both pairs of coils separated by 1.5 mm. In all setups, the input power was set at 845 mW. Using these three measurements, a breakdown of the energy dissipated in the system can be made. This is shown in Figure 14.

The data show that the majority of losses occur in the inverter and link. With the current data, it is not possible to ascertain the losses in these two elements separately; however, it is likely that they are mostly caused by inefficiencies in the inverter. This indicates that the ZVS condition might not always be met during operation, thus, resulting in a significant drop in efficiency. This can be improved by lowering the value of the resonant inductor $L_{shunt}$ and by adjusting the clock signal duty cycle.

To test the system performance while transmitting data, a placeholder data protocol was implemented. This allowed data communication at 8.5 kbits/s from the slave to the master with acknowledgments. While transmitting data, the power efficiency dropped to 9%, thus, resulting in 62 mW being available at the load.

### 3.3. Coupling Correction

In the previous section, we suggested that the phase of the current compared to the voltage could be used to optimize $C_{comp}$ to changing coupling conditions. In order to test this hypothesis, the output of the phase detector and the system efficiency were measured at various DC loads and values of $C_{comp}$. This produced the plots shown in Figure 15.

The first thing that can be noted is that there is a clear correlation between the detected phase and the system efficiency. This correlation is most pronounced at lower load impedances. There appears to be a correlation between the measured phase and the load impedance itself as well. This would have to be corrected for in order to use the current phase for the self-optimization of a system.

Secondly, the plots show two values of $C_{comp}$ that result in peak in efficiency. According to (Equation 6), a peak is expected at 140 pF, which can be observed in the measurements. The other peak is not explainable using linear circuit theory but is likely a result of the complex non-linear nature of the rectifier and carrier generator. However, since the phase detector is also capable of detecting this second peak, it will not affect the system in a significant way.

### 3.4. Future Work

The system built in this work serves as a proof of principle. Improvements can be made to increase both the performance and practicality of the system. The next step would be to implement the coils inside a piece of fabric using conductive thread and to take measurements to asses how this influences the behavior of the link. Investigating the effects caused by the finite conductivity of the wire and flexing of the coils are of particular relevance.

Once the integrated system is verified, the supporting electronics can be further optimized and miniaturized. In particular, the carrier generator can be made more efficient by more strictly enforcing the ZVS condition.

The data protocol used in our system was intended as a placeholder. A well-designed data protocol would allow for half-duplex communication at higher data rates than is currently achieved. Research could be performed to find the optimal trade-off between the data speed and power transfer rate. Alternatively, by using a different approach for the modulation of data onto the signal, the trade-off between power and data could be eliminated entirely.

Comparing with the system presented in [29], our system represents a nearly eight-fold increase in power transfer. The demonstrated 8.5 kbit/s data transfer is an improvement of nearly 80 times. It should however be kept in mind that the system in [29] uses embroidered coils, whereas our system currently uses coils made of rigid materials. Nevertheless, our system is specifically designed with the use of flexible coils in mind, and their use should not significantly hamper performance.

## 4. Conclusions

This work showed that sending data and power across two air gaps using inductive coupling was both effective and feasible. We demonstrated analytically that the double inductive link behaved the same as a conventional single link with an applied transformation. Additionally, we demonstrated how a system consisting of small self-inductance coils was more resilient to changes in conditions than was a system with larger coils. The measurements with the prototype demonstrated that it is possible to estimate and correct for efficiency loss due to changes in the coupling between the coils.

Furthermore, the current–voltage phase relation was shown to be a good estimator for the system efficiency. It was proven that the system was able to transfer 62 mW of usable DC power, whilst simultaneously transmitting data at a rate of 8.5 kbit/s. The hardware itself was demonstrated to support data rates of up to 240 kbit/s. These results represent a significant improvement over previously published works.

## Figures and Tables

**Figure 1 sensors-23-02943-f001:**
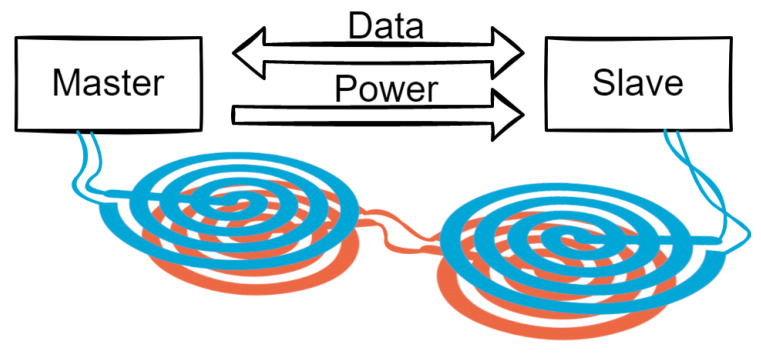
Illustration of the system.

**Figure 2 sensors-23-02943-f002:**
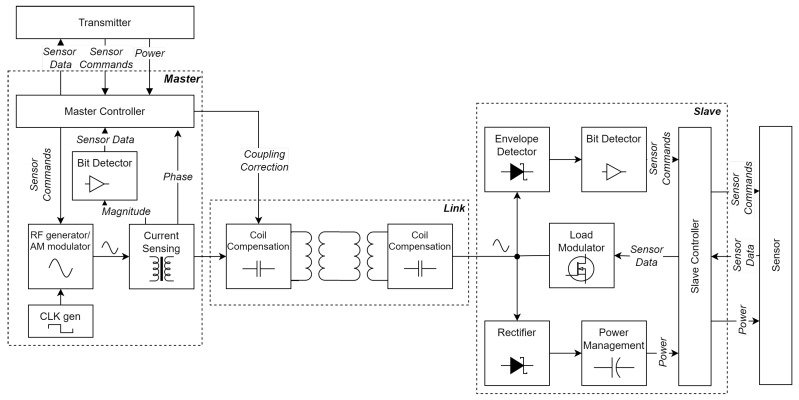
Block diagram of all the subsystems.

**Figure 3 sensors-23-02943-f003:**
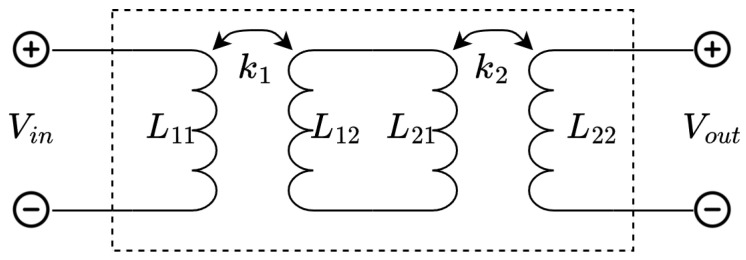
Circuit used for two-port modeling of the double link.

**Figure 4 sensors-23-02943-f004:**
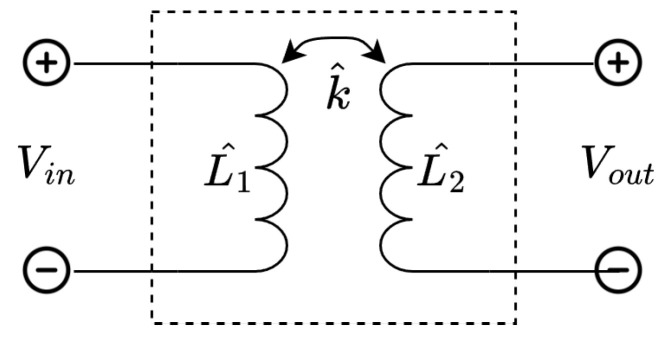
Circuit used for two-port modeling of a single link.

**Figure 5 sensors-23-02943-f005:**
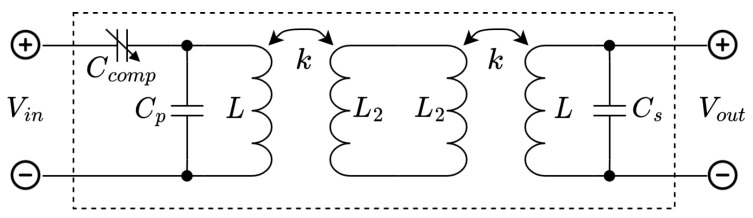
Proposed three capacitor compensation network.

**Figure 6 sensors-23-02943-f006:**
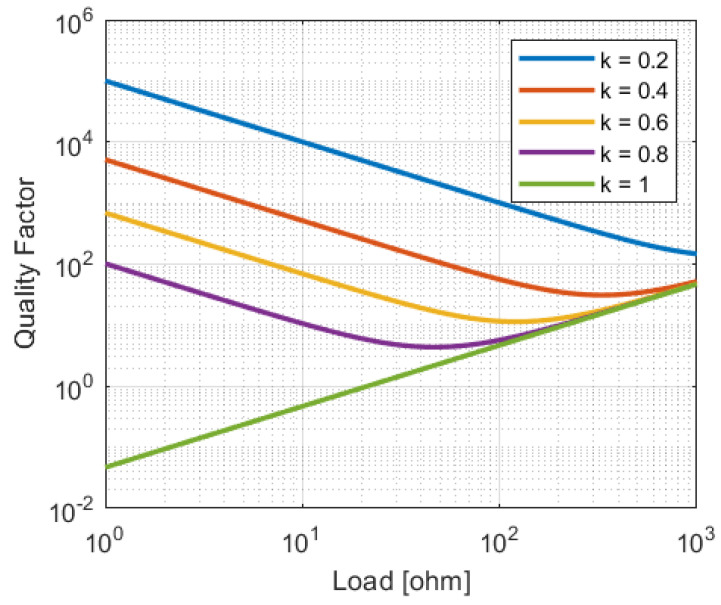
System *Q* for various loads and coupling factors (k). L=500 nH and ω=2π·13.56 MHz.

**Figure 7 sensors-23-02943-f007:**
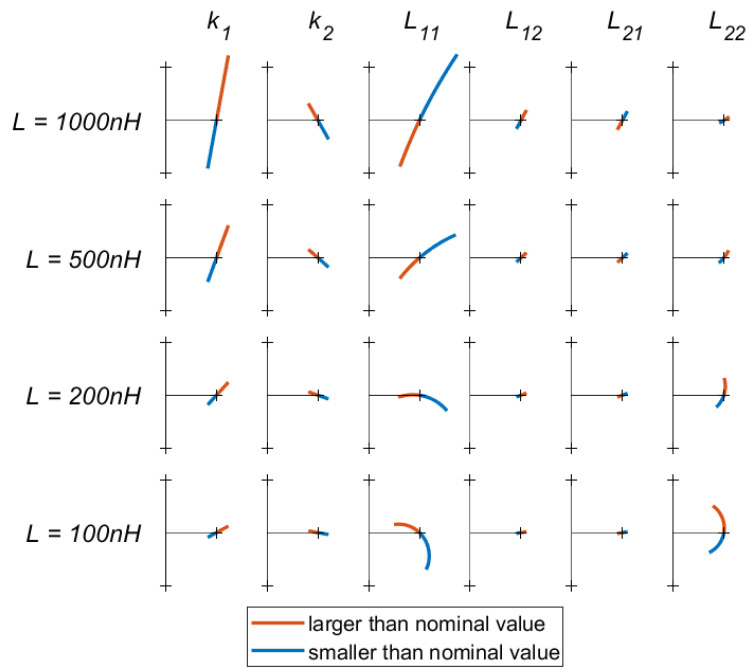
A grid of Nyquist plots showing how the real part (x-axis) and imaginary part (y-axis) of the input impedance of the link change as one of the link parameters ($k_{1}$, $k_{2}$, and $L_{11}$–$L_{22}$) is varied from −10% to +10% of its nominal value. Each column of plots corresponds to a different parameter, whereas the rows indicate different nominal values for *L*. A longer line indicates a greater sensitivity to the selected parameter. The tick-marks on the axis are placed at a 50 Ω distance from the origin. Each simulation was performed at a frequency of 13.56 MHz with a 50 Ω load and a nominal coupling factor of 0.7.

**Figure 8 sensors-23-02943-f008:**
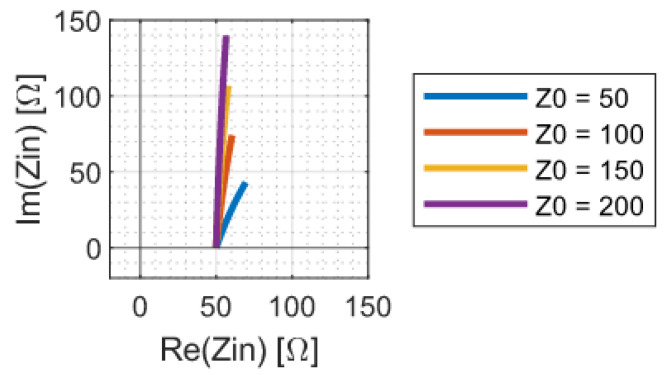
Nyquist plot of the link input impedance as the length of the transmission line between $L_{12}$ and $L_{21}$ increases from 0 to 0.5 m. The various traces correspond to different characteristic transmission line impedances. The transmission line used in the simulation has a wave propagation speed of 0.6c and is terminated in a 50 Ω load.

**Figure 9 sensors-23-02943-f009:**
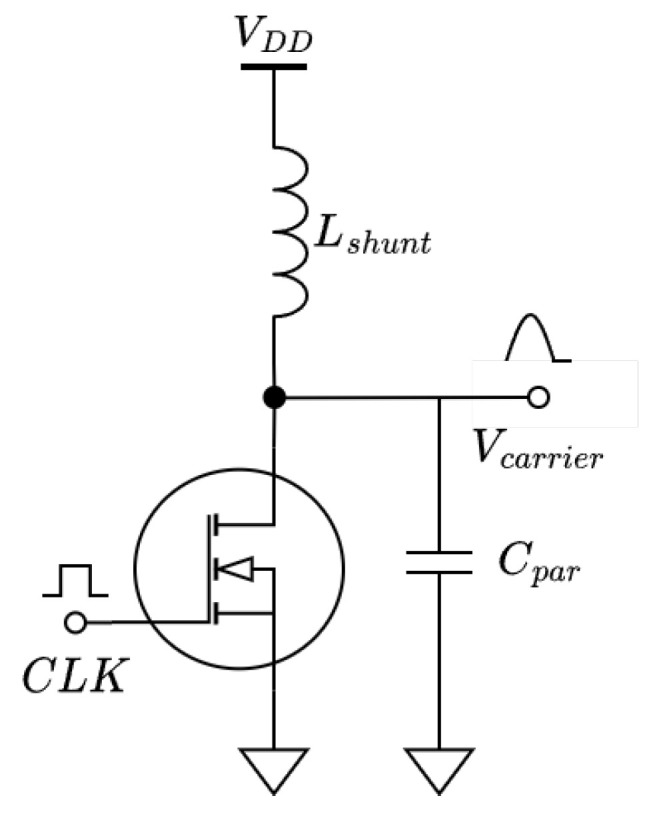
Modified class-E amplifier.

**Figure 10 sensors-23-02943-f010:**
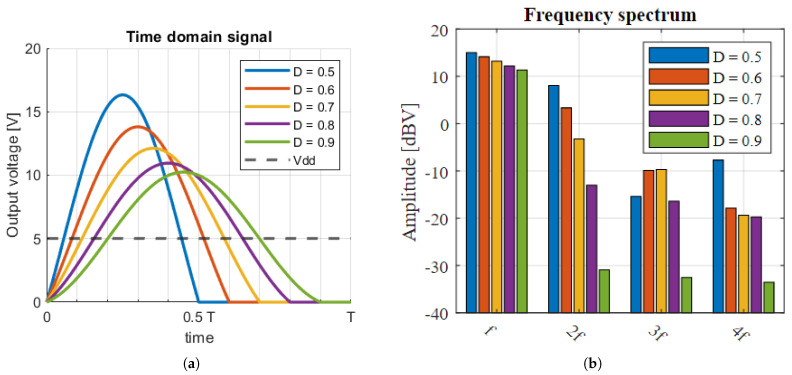
(**a**) Simulated time-domain waveforms of the output signal of the modified class-E amplifier. (**b**) Spectral use of the output signal.

**Figure 11 sensors-23-02943-f011:**
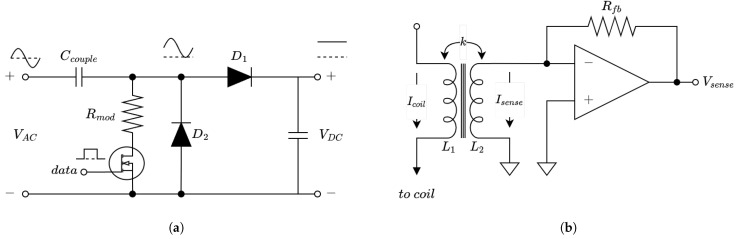
(**a**) Rectifier and load modulator. (**b**) Current-sensing network.

**Figure 12 sensors-23-02943-f012:**
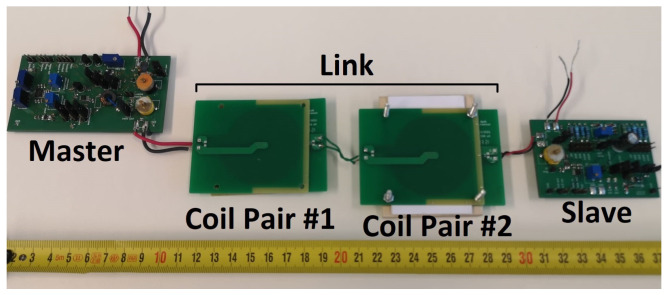
Image of the prototype. In this picture, coil pair #2 is placed in a holder with a fixed spacing, and coil pair #1 is loose.

**Figure 13 sensors-23-02943-f013:**
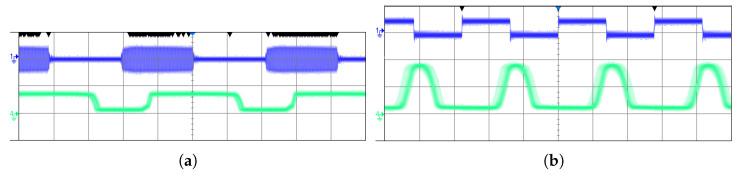
Signal waveforms during square wave communication. (**a**) Master-to-slave communication at 120 kHz. The top signal is the modulated carrier sent by the master. The bottom signal is the demodulated signal output by the slave. The vertical scale is 5 V/div. (**b**) Slave to master communication at 180 kHz. The top signal is the data stream presented to the input of the slave. The bottom signal is the demodulated signal output by the master. The vertical scale is 2 V/div.

**Figure 14 sensors-23-02943-f014:**
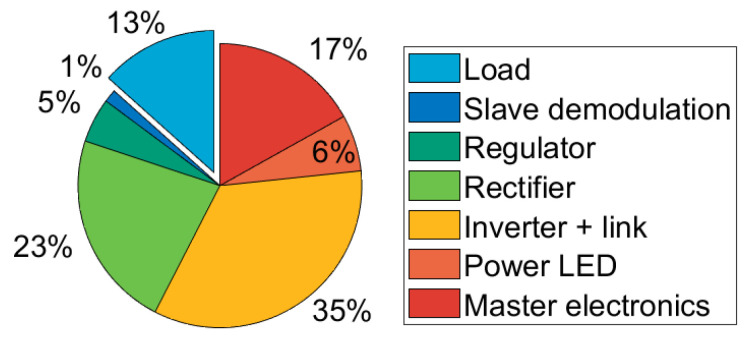
Estimate of the power lost in the various subsystems.

**Figure 15 sensors-23-02943-f015:**
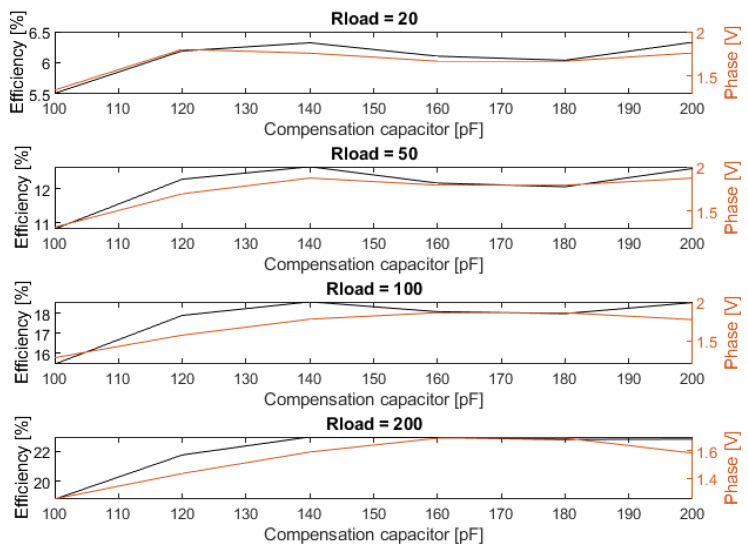
Measured DC efficiency and phase detector output for various DC loads at a coil distance of 1.5 mm.

**Table 1 sensors-23-02943-t001:** Common values for primary compensation capacitors and their resulting transmission matrices at resonance ($\omega =\omega_{0}$).

Topology	$C_{p}$	$C_{s}$	$\left[\begin{array}{c} T \end{array}\right]$
SS	$\frac{1}{\omega_{0}^{2}\hat{L}}$	$\frac{1}{\omega_{0}^{2}\hat{L}}$	$\left[\begin{array}{cc} 0 & \frac{\omega_{0}\hat{k}\hat{L}}{j}\\ \frac{j}{\omega_{0}\hat{k}\hat{L}} & 0 \end{array}\right]$
SP	$\frac{\hat{1}}{\omega_{0}^{2}\hat{L}\left(1-{\hat{k}}^{2}\right)}$	$\frac{1}{\omega_{0}^{2}\hat{L}}$	$\left[\begin{array}{cc} \hat{k} & 0\\ 0 & \frac{1}{\hat{k}} \end{array}\right]$
PP	$\frac{\left(1-{\hat{k}}^{2}\right)\hat{L}}{\omega_{0}^{2}{\hat{L}}^{2}{\left(1-{\hat{k}}^{2}\right)}^{2}+{\hat{k}}^{4}Z_{l}^{2}}$	$\frac{1}{\omega_{0}^{2}\hat{L}}$	$\left[\begin{array}{cc} \hat{k} & \frac{\hat{L}\omega_{0}\left(1-{\hat{k}}^{2}\right)}{\hat{k}}\\ \frac{j\omega_{0}\hat{L}\hat{k}\left(1-{\hat{k}}^{2}\right)}{\omega_{0}^{2}{\hat{L}}^{2}{\left(1-{\hat{k}}^{2}\right)}^{2}+Z_{l}{\hat{k}}^{4}} & \frac{Z_{l}{\hat{k}}^{4}}{\omega_{0}^{2}{\hat{L}}^{2}{\left(1-{\hat{k}}^{2}\right)}^{2}+Z_{l}{\hat{k}}^{4}} \end{array}\right]$
PS	$\frac{Z_{l}^{2}}{\omega_{0}^{2}\hat{L}Z_{l}^{2}+\omega_{0}^{3}\hat{L}{\hat{k}}^{4}}$	$\frac{1}{\omega_{0}^{2}\hat{L}}$	$\left[\begin{array}{cc} \frac{1}{\hat{k}} & \frac{\omega_{0}\hat{k}\hat{L}}{j}\\ \frac{\hat{L}{\hat{k}}^{3}\omega_{0}}{j{\hat{L}}^{2}{\hat{k}}^{4}\omega_{0}^{2}+jZ_{l}} & \frac{\hat{k}Z_{l}}{{\hat{L}}^{2}{\hat{k}}^{4}\omega_{0}^{2}+Z_{l}} \end{array}\right]$

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
