# Peer review of "Adaptive Impedance Matching Network for Contactless Power and Data Transfer in E-Textiles"

_sensors, 2023, doi:10.3390/s23062943_

Round 1

Reviewer 1 Report

Kindly go through my comments

Author Response

Dear Reviewer,

Thank you very much for your valuable comments. Please, find our point-by-point response to the comments below. The adjustments are highlighted in red in the pdf. Text that is to be removed is shown crossed out. Changed line numbers are also provided where relevant.

Comments:

  1. The method section in the abstract is not clearly indicated

To make the abstract easy-to-read, we presented the approach and method together with the associated results. We believe the current abstract is a good representation of the content of the article. As far as we are aware, MDPI does not require various sections within the abstract to be explicitly labeled.

  1. In the literature, you need to cite very recent articles which relates to your title such as: https://www.mdpi.com/1996-1944/14/17/5113; https://www.mdpi.com/2072-666X/11/2/209; https://www.mdpi.com/1996-1944/15/1/272; https://doi.org/10.1002/adma.201901958 etc

Some more context on mechanical bonding methods used in e-textiles has been added as an additional paragraph in the introduction. This includes references to some of the papers you mentioned.

  1. If Figure 2 is not your own, it needs reference and as well as permission

Figure 2 is made by us, it was used previously in the master thesis this work is based on. This master thesis has been added as a reference in the introduction.

  1. In the introduction, you did not discuss issues related to your title

The title has been changed from “Contactless Power and Data Transfer for use in E-Textiles” to “An Adaptive Impedance Matching Network for Enabling Contactless Power & Data Transfer in E-Textiles“, which hopefully better reflects the content of the article.

  1. Why do you choose Link Design?

The chosen topology for the link and its compensation network directly mirrors the impedance at one end of the link to the other. This simplifies the design of the other components in the system, as it can be assumed that any voltage and current on one side of the link is also present on the other side.

Furthermore, it allows a single variable capacitor to tune the network for a wide range of coupling factors. This is mentioned in line 149 of the manuscript.

Aside from these benefits, it also does not require any components to be placed in between the two pairs of coupled coils. This should greatly increase ease of manufacturing, lifetime, and recyclability of any garment that incorporates this system.

  1. You chose rigid coils, is it good for wearable textiles related to comfort?

Flexible coils would indeed be more comfortable and are very much desirable for further iterations of this project. However, because flexible coils are more difficult to produce and to perform consistent measurements with, they were omitted in this research. Our goal was to first verify the validity of the methods used, before expanding to a system made from flexible materials. This is indicated in lines 226 and 371 of the article.

  1. Where is the method section? Not indicated clearly

The manuscript has been updated, it now clearly indicates the methods section.

  1. Some results in the method section seems to be referred carefully

We are not sure what is meant with this comment.

  1. Typographical and grammar errors must be carefully checked

The text was carefully checked for errors

  1. The title and the body don’t much

See point 4.

  1. Any estimation on the lose on the inverter and link

The methods we used do not allow separate measuring of the losses in the link and in the inverter. However, it is expected that a large majority of these losses occur in the inverter. This is indicated in line 340 of the manuscript.

Reviewer 2 Report

In this study, the authors aim to increase the user experience and mechanical reliability of these connections by foregoing conventional galvanic connections in favour of inductively coupled coils. Although the topic is interesting there are some shortcomings which are given below:

The extended version of this study is also available on the internet as master thesis. The authors should cite this thesis and highlighted the differences between this published thesis. On the other hand, the authors should compare their results with more published papers, not only Lin et al. who obtained 112 bits/s transfer rate.  

The authors should discuss that which part of the body of this hardware will be placed. Furthermore, according to the placed part what will be the noise level comes from body movement, temperature, humidity etc.

The authors should discuss the limitation of the proposed method.

Page 5 line 148 the this is not a new paragraph.

There are one-sentence paragraphs in the manuscript. This is not suitable.

The Picture Figure 12 should be taken as top view. Side view is not well-presentation.

Author Response

Dear Reviewer,

Thank you very much for your valuable comments. Please, find our point-by-point response to the comments below. The adjustments are highlighted in red in the pdf. Text that is to be removed is shown crossed out. Changed line numbers are also provided where relevant.

Comments:

In this study, the authors aim to increase the user experience and mechanical reliability of these connections by foregoing conventional galvanic connections in favour of inductively coupled coils. Although the topic is interesting there are some shortcomings which are given below:

  1. The extended version of this study is also available on the internet as a master thesis. The authors should cite this thesis and highlighted the differences between this published thesis. On the other hand, the authors should compare their results with more published papers, not only Lin et al. who obtained 112 bits/s transfer rate.

On line 78 a paragraph has been added which refers to the masters thesis. The main difference between the thesis and this publication is further analysis of the effect changing coil conditions have on the system.

Comparing our results with more published papers has proven difficult. As mentioned in the paper, there are few published sources on the achieved data and power transfer rates of similar systems. The paper by Lin et al. was the only one found suitable for a direct comparison. However, to aid the reader in judging the performance of the system, a new paragraph was added on line 46. This paragraph outlines the achieved results using similar techniques in different applications.

  1. The authors should discuss that which part of the body of this hardware will be placed. Furthermore, according to the placed part what will be the noise level comes from body movement, temperature, humidity etc.

In principle the systems can be placed anywhere on the body where it is comfortable to wear. Flex coils should be placed at logical positions where bending is limited, which still comprises most of the body. To emphasize this point, lines 48 and 73 have been added to the introduction.

  1. The authors should discuss the limitation of the proposed method.

In its current implementation there is a trade-off between the transfer of energy and the transfer of data. This is due to the use of OOK (On-Off Keying) for data transmission. Using a different modulation method makes demodulation more difficult, but would remove this trade-off. This has been further highlighted in the discussion section.

Of course the data and power transfer rates of our system will never be as good as when a direct wired connection is used, but the types of sensor networks this system is intended for generally will not require such high data and power transfer capabilities.

  1. Page 5 line 148 the this is not a new paragraph.

There was an error in the Latex code where after each equation the text would start with an indent, indicating a new paragraph where it was not intended. This has now been fixed.

  1. There are one-sentence paragraphs in the manuscript. This is not suitable.

See point 4.

  1. The Picture Figure 12 should be taken as top view. Side view is not well-presentation.

The picture has been replaced by an improved version.

Round 2

Reviewer 1 Report

Ok

Reviewer 2 Report

The authors addressed all the points. I believe that this version is acceptable.